# Characterization of a Novel Viral Interleukin 8 (vIL-8) Splice Variant Encoded by Marek’s Disease Virus

**DOI:** 10.3390/microorganisms9071475

**Published:** 2021-07-09

**Authors:** Yu You, Ibrahim T. Hagag, Ahmed Kheimar, Luca D. Bertzbach, Benedikt B. Kaufer

**Affiliations:** 1Institute of Virology, Freie Universität Berlin, 14163 Berlin, Germany; yuyou@zedat.fu-berlin.de (Y.Y.); ibrahim.hagag@fu-berlin.de (I.T.H.); ahmed1985@zedat.fu-berlin.de (A.K.); 2Department of Virology, Faculty of Veterinary Medicine, Zagazig University, Zagazig 44511, Egypt; 3Department of Poultry Diseases, Faculty of Veterinary Medicine, Sohag University, Sohag 82424, Egypt; 4Department of Viral Transformation, Leibniz Institute for Experimental Virology (HPI), 20251 Hamburg, Germany

**Keywords:** Marek’s disease virus, viral chemokine, vIL-8, CXCL13, alternative RNA splicing, splice acceptor site, pathogenesis, tumorigenesis, chickens

## Abstract

Marek’s disease virus (MDV) is a highly cell-associated oncogenic alphaherpesvirus that causes lymphomas in various organs in chickens. Like other herpesviruses, MDV has a large and complex double-stranded DNA genome. A number of viral transcripts are generated by alternative splicing, a process that drastically extends the coding capacity of the MDV genome. One of the spliced genes encoded by MDV is the viral interleukin 8 (vIL-8), a CXC chemokine that facilitates the recruitment of MDV target cells and thereby plays an important role in MDV pathogenesis and tumorigenesis. We recently identified a novel vIL-8 exon (vIL-8-E3′) by RNA-seq; however, it remained elusive whether the protein containing the vIL-8-E3′ is expressed and what role it may play in MDV replication and/or pathogenesis. To address these questions, we first generated recombinant MDV harboring a tag that allows identification of the spliced vIL-8-E3′ protein, revealing that it is indeed expressed. We subsequently generated knockout viruses and could demonstrate that the vIL-8-E3′ protein is dispensable for MDV replication as well as secretion of the functional vIL-8 chemokine. Finally, infection of chickens with this vIL-8-E3′ knockout virus revealed that the protein is not important for MDV replication and pathogenesis in vivo. Taken together, our study provides novel insights into the splice forms of the CXC chemokine of this highly oncogenic alphaherpesvirus.

## 1. Introduction

Marek’s disease virus (MDV) is a highly contagious and strictly cell-associated alphaherpesvirus that causes a deadly lymphoproliferative disease in chickens [1]. The virus causes substantial economic losses in poultry production worldwide and causes mortality rates of up to 100% in unvaccinated chickens [2,3,4,5]. Therefore, billions of chickens are vaccinated every year. Aside from the deadly lymphomas, the virus also induces immunosuppression and severe neurological symptoms. MDV has a double-stranded DNA genome of approximately 180 kilo base pairs, consisting of a long and a short unique sequence region (U_L_ and U_S_), which are flanked by terminal repeats (TR_L_ and TR_S_) and internal repeats (IR_L_ and IR_S_, Figure 1A) [2]. The MDV genome encodes more than 100 open reading frames (ORFs), which are involved in various processes including replication, immune evasion, pathogenesis, and virus-induced tumor formation [6].

MDV encodes a CXC chemokine that was initially named viral interleukin 8 (vIL-8) but is most closely related to chicken CXCL13 L1 [7]. vIL-8 is encoded in the TR_L_ and IR_L_ of the MDV genome (*MDV003*/*MDV078*) and expressed with true late kinetics [7,8]. vIL-8 is secreted from MDV-infected cells and recruits B cells and CD4^+^ CD25^+^ T cells, which are target cells for virus replication and MDV-induced transformation, respectively [9,10]. The vIL-8 gene consists of two introns and three exons, which are spliced to produce a ~0.7 kilo base transcript. Exon I serves as a short signal peptide [11], while exons II and III are spliced to generate the secreted chemokine. Intriguingly, exons II and III are also spliced to upstream genes such as the major MDV oncogene *meq*, *RLORF4*, and *RLORF5a* [12]. Deletion of the vIL-8 gene from the MDV genome (almost) completely abrogated disease and tumor formation [8,11]. In contrast, abrogation of vIL-8 chemokine expression by mutating its start codon or deleting its exon I only reduced disease and tumor incidence by about 60% [9,12], suggesting that alternative splice forms likely play a role in MDV pathogenesis.

Comprehensive transcriptome analyses of MDV-infected B cells and chicken embryo cells (CECs) recently revealed a novel alternative vIL-8 splice junction within intron II [6,13]. This splice event would result in a novel exon 3 (E3′) containing the last 16 base pairs (bp) of intron II and a stop codon. While the splice variant was clearly detectable in the transcriptome of B cells and CECs, it remained unknown whether this spliced transcript encodes a protein. 

In this study, we set out to investigate whether this novel vIL-8 splice variant is expressed on the protein level and assessed its role in MDV replication and pathogenesis. We generated recombinant viruses that either have a FLAG-tagged E3′ or lack its expression by mutating the splice acceptor site. We could demonstrate that the novel splice form is expressed as a protein. This protein is dispensable for virus replication, and its absence does not affect expression of the secreted vIL-8 chemokine. Only minor effects were observed in MDV pathogenesis and tumor formation. Our study represents the first characterization of this novel vIL-8 splice form encoded by this highly oncogenic avian herpesvirus.

## 2. Materials and Methods

### 2.1. Ethics Statement

All animal work was conducted according to relevant international and national guidelines for the care and the humane use of animals and was approved by the LAGeSo (Landesamt für Gesundheit und Soziales) Berlin, Germany (approval number G0294-17, approval date 16.1.2018).

### 2.2. Cells

CECs were isolated from embryonated VALO specific pathogen-free eggs (VALO BioMedia GmbH; Osterholz-Scharmbeck, Germany) as described previously [14]. CECs were cultured in Eagle’s minimum essential medium (PAN Biotech; Aidenbach, Germany), supplemented with 1% to 10% fetal bovine serum (PAN Biotech) and antibiotics (100 U/mL penicillin and 100 µg/mL streptomycin; AppliChem; Darmstadt, Germany) at 37 °C and 5% CO_2_.

### 2.3. Viruses

All recombinant viruses were generated based on a previously generated bacterial artificial chromosome (BAC) clone of the very virulent RB1B strain in which most of the internal repeat regions were deleted (RB1B-ΔIR_LS-HR_, GenBank number MT955328) [15]. This deletion is rapidly restored upon reconstitution and facilitates a rapid manipulation of the repeat regions using two-step Red-mediated mutagenesis [15,16,17]. In addition, a pTK-eGFP cassette in the mini-F allowed the detection of infected cells in vitro, while the mini-F was removed for in vivo studies [18]. All primers used for the mutagenesis are listed in Table 1. The following viruses were generated in our study: vE3′-FLAG contains a FLAG tag with a glycine-serine (GS) linker at 89 bp of the vIL-8 intron II, which is thereby only encoded in the novel vIL-8-E3′ and not the previously known vIL-8 exons; vΔE3′ and vΔE3′-FLAG contain a point mutation (G to A) in the novel splice acceptor site at 82 bp of intron II to abrogate splicing and expression of this putative novel protein (Figure 1A). All recombinant viruses were confirmed by restriction fragment length polymorphism (RFLP), PCR, Sanger sequencing, and Illumina MiSeq sequencing with more than 1000-fold coverage to ensure that the entire viral genome is correct [19]. In addition, we used a previously generated mutant virus that lacks the expression of the vIL-8 chemokine (vΔMetvIL-8) due to the mutation of its start codon [9].

### 2.4. Western Blotting

To investigate the expression of the novel vIL-8 splice variant and vIL-8 secretion, Western blot analyses were performed as described previously [20]. Briefly, CECs were infected with 10,000 plaque-forming units (pfu) of the indicated viruses. Cells or supernatants were harvested at 5 days post infection (dpi). Samples were separated by SDS-PAGE and then transferred to a polyvinylidene difluoride membrane (Carl Roth; Karlsruhe, Germany) using the Biometra semi-dry blotting system (Biometra; Göttingen, Germany). Subsequently, the membranes were blocked with 5% milk in phosphate-buffered saline (PBS) and incubated overnight at 4 °C with a mouse monoclonal α-FLAG tag antibody (1:1000; ABM; Richmond, Canada), the rabbit polyclonal anti-vIL-8 antibody [8], or the mouse monoclonal anti-gC antibody [9,21], respectively. After three washes with PBST (PBS containing 0.05% Tween 20), the membranes were incubated for one hour at room temperature with horseradish peroxidase (HRP)-conjugated goat anti-mouse or anti-rabbit antibodies (1:10,000; Cell Signaling; Danvers, MA, USA). Finally, membranes were visualized using enhanced chemiluminescence (ECL) plus substrate (Thermo Fisher Scientific; Waltham, MA, USA), and protein signals were visualized with the Chemi-Smart 5100 detection system (Peqlab; Erlangen, Germany).

### 2.5. Indirect Immunofluorescence

To investigate the protein expression of the novel splice form, indirect immunofluorescence analysis (IFA) was performed as described previously [20]. Briefly, CECs were infected with 100 pfu, fixed at 5 dpi with 4% paraformaldehyde, and blocked with 3% BSA for 30 min. Subsequently, cells were stained with the mouse monoclonal α-FLAG tag antibody (1:500; ABM; Richmond, Canada) and incubated for 45 min at room temperature. Cells were washed with PBS, probed with Alexa goat anti-mouse IgG (H + L) 568 antibody (1:1000; Invitrogen; Carlsbad, CA, USA), and incubated at room temperature for 45 min. After three washes with PBS, cells were stained with DAPI stain (5 μg/mL) in PBS. Cells were examined and images captured with an AxioVision microscope (Zeiss; Oberkochen, Germany).

### 2.6. Plaque Size Assays

To assess cell-to-cell spread of the recombinant viruses, we performed plaque size assays as previously described [15,22]. Briefly, one million CECs were infected with 100 pfu of the indicated viruses. At 5 dpi, images of randomly selected plaques (*n* = 50) were captured and analyzed using the ImageJ software (NIH; Madison, WI, USA). Plaque diameters were measured and compared with the controls.

### 2.7. Multi-Step Growth Kinetics

Replication properties of the recombinant viruses were assessed by quantitative PCR (qPCR)-based multi-step growth kinetics as previously described [15,22]. Briefly, one million CECs were infected with 100 pfu of the indicated viruses. Cells were harvested at the indicated time points over the course of five days, and viral DNA was extracted using the RTP DNA/RNA Virus Mini kit (Stratec; Berlin, Germany). MDV genome copies of three independent experiments were evaluated by qPCR. Primers and probes specific to the MDV-infected cell protein 4 (ICP4) and chicken inducible nitric oxide synthase (iNOS) are listed in Table 1. Virus genome copies were normalized against the chicken iNOS gene.

### 2.8. In Vivo Experiment

To investigate the role of vIL-8-E3′ in MDV replication and pathogenesis in vivo, one-day-old specific pathogen-free (SPF) VALO chickens (VALO BioMedia) were randomly distributed into three groups and housed separately. The chickens of each group were infected subcutaneously with 4000 pfu of either the wild type (*n =* 10), vΔE3′ (*n =* 25), or vMetvIL-8 (*n =* 24). In addition, age-matched naïve chickens (WT (*n =* 11), vΔE3′ (*n =* 10), and vMetvIL-8 (*n =* 10)) were co-housed with experimentally infected chickens to assess the natural transmission of the respective viruses. The experiment was performed in a blinded manner to eliminate any subjectivity. Chickens were monitored daily for the onset of clinical symptoms. Once clinical signs were detected or at termination of the experiment (at 91 dpi), chickens were humanely euthanized and examined for gross tumor lesions.

### 2.9. Virus Quantification in Blood and Feather Follicles

To assess virus replication in vivo, whole blood samples were collected from infected animals at 4, 7, 10, 14, 21, and 28 dpi (*n =* 8) as well as contact animals at 21, 28, and 35 dpi (*n =* 8). DNA was isolated from all blood samples using the NucleoSpin 96 Blood Core Kit (Macherey-Nagel; Düren, Germany) according to the manufacturer’s instructions. To evaluate the efficiency of the virus delivery to and replication in the feather follicle epithelium (FFE), proximal ends of each feather containing the feather pulp were collected from infected birds at 14, 21, 28, 35, and 42 dpi as described previously [19,23]. DNA was extracted through treatment of the feather pulp with proteinase K at 55 °C overnight, followed by phenol:chloroform:isoamyl alcohol extraction and ethanol precipitation as described previously [24]. MDV genome copies were measured by qPCR as described above.

### 2.10. Statistical Analyses

Statistical analyses were performed using GraphPad Prism v8 (GraphPad Software, Inc.; San Diego, CA, USA). The multi-step growth kinetics were analyzed with the Kruskal–Wallis test. Analysis for plaque size assays was performed by a one-way analysis of variance (ANOVA). Data on the number of MDV genome copies in whole blood and feather samples were analyzed using the Kruskal–Wallis test. Disease incidence curves were analyzed using the log-rank test (Mantel–Cox test); Fisher’s exact test was used for tumor incidences and tumor distribution with corrections on multiple comparisons. Data were considered significant if *p* ≤ 0.05.

## 3. Results

### 3.1. Detection of Protein Expression of the Novel vIL-8 Splice Variant

To determine whether the novel splice variant is expressed as a protein, we generated a recombinant MDV (vE3′-FLAG) containing a FLAG tag in frame with the putative protein. The FLAG tag was inserted within intron II downstream of the novel acceptor splice site A19′ (Figure 1A). We infected CECs with E3′-FLAG or the parental virus and investigated the expression of the putative protein. Western blot analysis revealed that the 8.8 kilo Daltons (kDa) protein was efficiently expressed and confirmed the existence of the putative protein containing the novel exon 3 (E3′) of vIL-8. The size is consistent with the protein encoded by the alternatively spliced vIL-8 transcripts fusing exons I, II, E3′ and the FLAG tag. 

To validate the expression of the novel vIL-8 splice variant on the cellular level, we performed IFA. Expression of the FLAG-tagged protein was clearly detectable in vE3′-FLAG-infected cells, while it was not detected in cells infected with the parental virus (Figure 1C). Our data demonstrate that the E3′ exon indeed gives rise to an alternatively spliced vIL-8 protein; however, it remained elusive whether this splice form plays a role in MDV replication and pathogenesis. 

### 3.2. Abrogation of the Novel vIL-8 Splice Variant

Previous studies demonstrated that individual splice variants can be abrogated by mutating the splice acceptor sites without affecting other gene products of the respective gene [25,26]. To determine the role of the novel vIL-8 splice variant, we replaced the acceptor splice site (A19′) AG to AA to abrogate the expression of the novel isoform (vΔE3′). To confirm that the vIL-8-E3′ protein is indeed not expressed, we also generated a mutant virus (vΔE3′-FLAG) harboring the FLAG tag in the deletion mutant. IFA analysis revealed that the expression of the novel splice variant was indeed abrogated in vΔE3′-FLAG-infected cells (Figure 1C), while it was readily detectable with the mutant harboring only the FLAG tag. Our data highlight that the novel acceptor splice site A19′ is crucial for the expression of the novel vIL-8 splice variant.

Next, we assessed the role of the novel splice variant in viral replication and cell-to-cell spread using multi-step growth kinetics and plaque size assays. Plaque size assays revealed that abrogation of the novel splice variant did not significantly affect virus replication and spread in culture compared with the parental virus (Figure 2A). These data were confirmed by multi-step growth kinetics that also did not reveal any significant differences (Figure 2B). Furthermore, we analyzed whether secretion of the vIL-8 chemokine is affected due to the abrogation of the novel splice variant. We infected CECs with the respective viruses and harvested the supernatant. Western blot analyses revealed that the levels of the vIL-8 protein in the supernatant were comparable between the vΔE3′ and parental virus (Figure 2C), highlighting that vIL-8 secretion is not dependent on the novel splice variant.

### 3.3. Role of the Novel vIL-8 Splice Variant in MDV Pathogenesis and Tumorigenesis

To investigate whether the novel vIL-8 splice variant contributes to MDV replication, pathogenesis, and tumor formation, we infected one-day-old chickens subcutaneously with 4000 pfu of vΔE3′, vΔMetvIL-8, or wild type (WT) virus. The vΔMetvIL-8 mutant served as a reference as it lacks expression of the secreted vIL-8 chemokine, resulting in significantly reduced disease and tumor incidence, but allows splicing of all vIL-8 variants as published previously [9]. First, we determined whether abrogation of the novel vIL-8 splice variant affects MDV replication in vivo. We quantified viral genome copies in the blood of infected chickens by qPCR at various time points post infection. vΔE3′ replicated efficiently in infected chickens (Figure 3A), indicating that the novel splice variant is dispensable for lytic replication in vivo. Moreover, we measured the virus load in FFE of the infected animals and observed high viral levels of vΔE3′ that were comparable to WT, suggesting that the novel splice variant does not contribute to the transport to and replication in the skin (Figure 3B). In contrast, virus load in the blood and the skin was significantly reduced in vΔMetvIL-8-infected animals in the absence of vIL-8 secretion (Figure 3A,B), which is consistent with a previous study [9].

In addition, we monitored the chickens for clinical disease symptoms and tumor incidence over the course of the 91-day experiment. Only 33.3% of the vΔMetvIL-8-infected chickens developed MD and tumors (Figure 3C–E), which is consistent with previous findings [9]. However, vΔE3′ induced disease and tumor induction as efficiently as in WT infections (Figure 3C–E), suggesting that the novel splice variant does not contribute to MDV pathogenesis. Taken together, our data demonstrate that the absence of the novel vIL-8 splice variant does not affect MDV replication, pathogenesis, tumor formation, and dissemination upon experimental infection. 

Previous studies showed that vIL-8 is essential for MDV pathogenesis and tumor formation in animals infected via the natural route of infection. To determine whether the novel vIL-8 splice variant plays a role in these processes during the natural infection, we co-housed naïve chickens with the subcutaneously infected chickens. vΔE3′ was efficiently transmitted to the contact chickens and replicated comparably to the parental virus (Figure 4A), suggesting that the establishment of MDV infection was not altered in the absence of the novel vIL-8 splice variant. We also monitored disease and tumor incidence during the entire experiment. None of the vΔMetvIL-8-contact chickens developed disease, while vΔE3′ viruses caused disease as efficiently as the parental virus (Figure 4B). Similarly, the tumor incidence (Figure 4C) and distribution (Figure 4D) of the vΔE3′ group were not significantly different from the group that was infected with WT. Taken together, our data demonstrate that the novel vIL-8 splice variant is not essential for Marek’s disease establishment in naturally infected contact animals.

## 4. Discussion

Herpesviruses have large and complex genomes and encode for many viral proteins. Over the last few years, next-generation sequencing analyses have revealed a wealth of novel herpesvirus genes and splice products [27,28,29]. Thus, the complexity of herpesvirus gene expression is further increased by the fact that a substantial number of viral proteins are encoded from spliced transcripts and/or alternative splicing. Several spliced viral transcripts have been described for MDV in vivo and in vitro, e.g., for the *meq*, vIL-8, and gC genes [12,13,30,31]. Recent studies have indicated that a spliced transcript of vIL-8 and *meq* play a role in MDV pathogenesis [32]. Consequently, there is an immense need to better understand the role of splicing in the MDV lifecycle and in MDV pathogenesis. 

So far, various vIL-8 splice variants have been identified, but it remained unknown whether each of the splice variants encodes a functional protein [12]. Recent comprehensive MDV transcriptome analyses identified a novel vIL-8 splice junction, which was detected at about 13–40-fold lower levels compared with previously published vIL-8 transcripts [6,12,13]. This new splice site was predicted to lead to a novel ORF that could potentially also splice with other transcripts through its splice acceptor site. In this study, we set out to validate the protein expression of the novel vIL-8 isoform using FLAG-tagged mutants. By Western blot analysis, we confirmed the existence of the novel splice variant containing exon E3′. A previous study reported an alternatively spliced Meq/vIL-8 transcript, which contains the bZIP amino terminus of Meq spliced with exon 2 and this short novel exon [33]. Intriguingly, this splice form was not detected at the protein level by Western blotting in infected CECs.

Spliced gene products can regulate protein levels or activities [25,34]. In MDV research, a recent study showed that the lack of Meq/vIL-8 splicing enhances virus replication during the late phase in infected chickens [32]. Therefore, we first investigated the novel E3′ splice variant in the context of virus replication and protein secretion. By mutating the A19′ splice acceptor site, we could demonstrate that neither MDV replication nor vIL-8 secretion was altered (Figure 2), which is consistent with previous findings demonstrating that a complete vIL-8 deletion or an abrogation of vIL-8 secretion did not affect virus replication in vitro [9,12]. Furthermore, abrogation of the novel vIL-8 splice variants did not affect virus replication during the early or late phase of infection in vivo (Figure 3A and Figure 4A). In light of these data, we concluded that the novel vIL-8 splice variant is dispensable for viral replication.

Complete deletion of vIL-8 severely impaired tumor incidence by more than 90% [8,11], while abrogation of the secreted chemokine without affecting the other splice variants still caused disease and tumors in about one third of the experimentally infected chickens (Figure 3C,D) [9]. Thus, the vIL-8 splice variants could potentially play roles in MDV-induced disease and tumor formation. Moreover, virus-encoded splice variants in general have been found to inhibit tumor suppressors, evade an immune response, and promote tumorigenesis [35,36]. To assess the effect of the novel splice variant on MDV pathogenesis, we infected one-day-old chickens with a virus that does not express the new vIL-8 splice variant. The recombinant virus (vΔE3′) showed disease and tumor incidences comparable to the WT virus (Figure 3C–E), suggesting that the novel vIL-8 splice variant is not essential for MDV pathogenesis and tumorigenesis.

Efficient horizontal transmission of MDV requires vIL8 secretion [9]. Similarly, previous studies revealed that all three gC splice variants are important for virus transmission [31]. Thus, we assessed the ability of these novel vIL-8 splice variants to spread to naïve co-housed contact chickens. We found that the vΔE3′ mutant was readily transmitted to contact chickens (Figure 4A) and observed comparable disease and tumor incidence to that in contact chickens infected with the WT via the natural route (Figure 4B–D), demonstrating that the novel vIL-8 splice variant is dispensable for transmission of MDV.

Interestingly, splicing of herpesviral genes is regulated by viral factors, such as the infected-cell protein 27 (ICP27) [37,38]. HSV-1 ICP27 predominantly transactivated unspliced gC mRNA and promoted the retention of an intron [39,40]. As described for HSV-1, MDV ICP27 can interact with splicing factors, which inhibits mRNA splicing of vIL-8 and the cellular chicken telomerase reverse transcriptase (chTERT) [41]. Upon ICP27 expression, unspliced vIL-8 transcripts were at low levels during MDV reactivation [41]. Thus, it would be very intriguing to further explore why MDV blocks vIL-8 splice variants during reactivation.

In summary, we investigated the role of a novel vIL-8 splice variant and could demonstrate that it is expressed as a protein but does not alter viral replication and disease outcomes. Although the novel vIL-8 splice variant is dispensable for MDV pathogenesis and tumorigenesis, our data provide a foundation for future studies on the diverse set of vIL-8 splice variants in the MDV lifecycle.

## Figures and Tables

**Figure 1 microorganisms-09-01475-f001:**
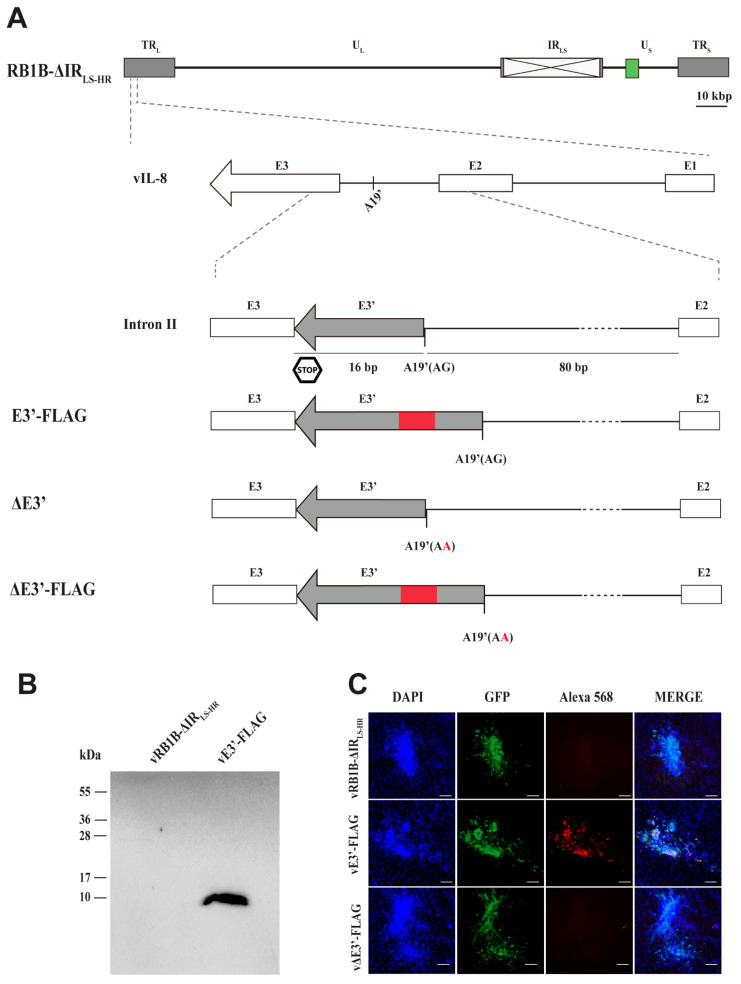
Identification and validation of the novel MDV vIL-8 splice variant containing E3′. (**A**) Schematic representation of the MDV genome (RB1B strain with deleted internal repeats, RB1B-ΔIR_LS-HR_) harboring the mini-F cassette with the pTK-eGFP cassette (green square), with a focus on the vIL-8 gene. The predicted splice acceptor site, potential exon, and stop codon are labeled with A19′, E3′, and stop, respectively. We inserted a FLAG tag (red square) at 89 bp downstream of the 5′ end of vIL-8 intron II (vE3′-FLAG). A point mutation (G to A) at the splice acceptor site was introduced to abrogate expression of the E3′splice variant (vΔE3′-FLAG and vΔE3′). (**B**) Western blot of the novel vIL-8 splice variant. CECs were infected with 10000 pfu and lysates harvested in RIPA I buffer. Lysates were subjected to SDS-PAGE and then immunoblotted with the mouse monoclonal α-FLAG tag antibody and a secondary anti-mouse IgM HRP-conjugated antibody. Cells infected with the parental virus RB1B-ΔIR_LS-HR_ were used as a negative control. (**C**) Detection of the novel vIL-8 splice variant by IFA. CECs were infected with 100 pfu of the indicated viruses, fixed at 5 dpi, and stained with a mouse monoclonal α-FLAG tag antibody (Alexa 568). Virus-infected cells express eGFP (green), and nuclei were visualized using DAPI (blue). The scale bars correspond to 100 μm.

**Figure 2 microorganisms-09-01475-f002:**
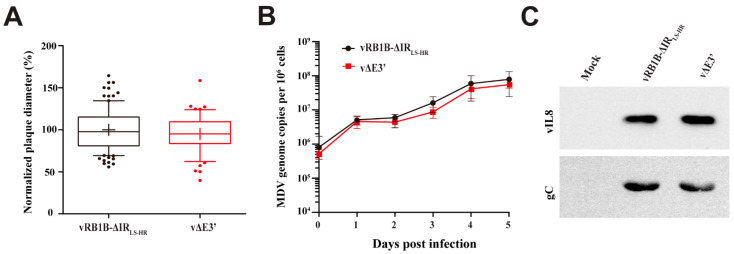
Characterization of vΔE3′ lacking the novel vIL-8 splice variant. (**A**) Plaque size assays of recombinant virus. Mean plaque diameters are shown as box plots with 5% to 95% confidence intervals (*n =* 50; *p* > 0.05, one-way ANOVA). (**B**) Viral replication was assessed by qPCR-based multi-step growth kinetics. Mean viral genome copies per one million cells with standard deviations are shown for the indicated viruses and different time points post infection (*p* > 0.05, Kruskal–Wallis test, *n =* 3). (**C**) Assessment of vIL-8 secretion by Western blotting. The supernatants of cells infected with the parental virus and vΔE3′ were harvested and membranes were incubated with vIL-8 and gC antibodies, respectively. The secreted MDV gC was used as a control for viral secretion products.

**Figure 3 microorganisms-09-01475-f003:**
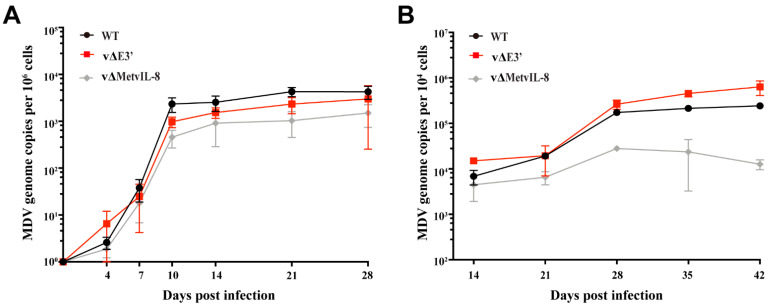
Experimental infection of chickens with vΔE3′ lacking the novel vIL-8 splice variant. (**A**) MDV genome copies were measured in whole blood samples of chickens infected with the indicated viruses by qPCR. Mean MDV genome copies per one million cells with standard deviations are shown for the indicated time points (*p* > 0.05, Kruskal–Wallis test). (**B**) MDV genome copies of the indicated viruses in feather tips of infected chickens (*p* > 0.05, Kruskal–Wallis test). (**C**) Disease incidences in chickens infected with the indicated recombinant viruses. Asterisks indicate significant differences in comparison with WT (*** *p* < 0.001, log-rank (Mantel–Cox) test). (**D**) Tumor incidence as the percentage of animals that developed tumors during the experiment. Asterisks indicate significant differences compared with WT (** *p* < 0.01; Fisher’s exact test). (**E**) Tumor distribution is shown as the number of tumorous organs in tumor-bearing animals with standard deviations (*p* > 0.05; Fisher’s exact test).

**Figure 4 microorganisms-09-01475-f004:**
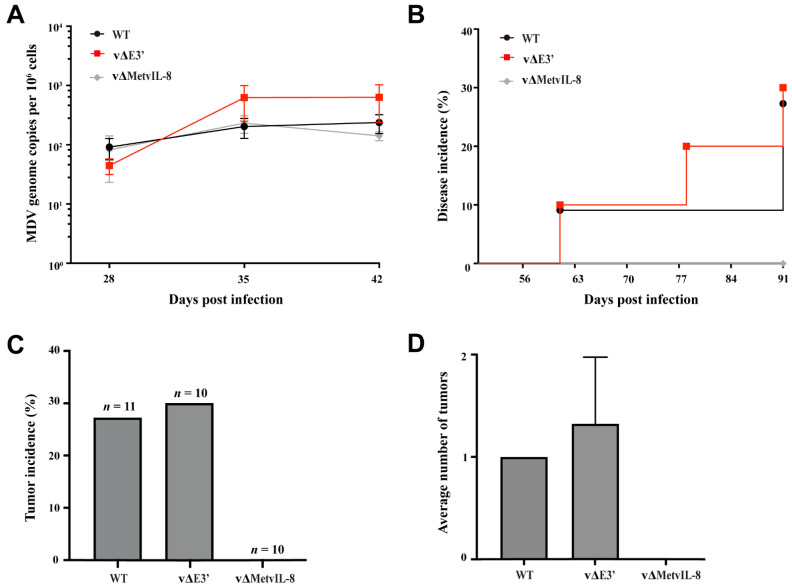
Pathogenesis and tumor incidence in contact chickens. (**A**) MDV genome copies were measured in whole blood samples from contact chickens by qPCR. Mean MDV genome copies per million cells with standard deviations are shown for the indicated time points (*p* > 0.05, Kruskal–Wallis test). (**B**) Disease incidence in naïve chickens infected with the indicated recombinant viruses (*p* > 0.05, log-rank (Mantel–Cox) test). (**C**,**D**) Tumor incidence and tumor distribution with standard deviations are shown for co-housed naïve chickens (*p* > 0.05, Fisher’s exact test).

**Table 1 microorganisms-09-01475-t001:** Oligonucleotide sequences used in this study.

Construct		Primer or Probe Sequence (5′–3′)
E3′-FLAG	For	**GTAGTGTCTGGCTGTAAAGCTAATTTGGTTAAGGTTTTCCG** *GCAGC GATTACAAGGATGACGACGATAAG* TAGGGATAACAGGGTAATCGATTT
	Rev	**ACATACCTTCCTGTTCTTCTTGAGAGCAAAGCTACAAAAG** *CTTAT CGTCGTCATCCTTGTAATCGCTGCC* GCCAGTGTTACAACCAATTAACC
vΔE3′-FLAG	For	**CTTCCTGTTCTTCTTGAGAGCAAAGCTACAAAAGGGAAAACTTTA ACCAAATTAGCTTTACAGCCAG** TAGGGATAACAGGGTAATCGATTT
	Rev	**CTTAGGTGTAGTGTCTGGCTGTAAAGCTAATTTGGTTAAAGTTTTC**CGCCAGTGTTACAACCAATTAACC
vΔE3′	For	**GCTACAAAAGCTTATCGTCGTCATCCTTGTAATCGGAAAACTTTA ACCAAATTAGCTTTACAGCCAG** TAGGGATAACAGGGTAATCGATTT
	Rev	**CTTAGGTGTAGTGTCTGGCTGTAAAGCTAATTTGGTTAAAG** **TTTTCC** GCCAGTGTTACAACCAATTAACC
vIL-8 sequencing	For	CCGTATCCCTGCTCCATCCAATAGC
	Rev	GGTCTCCAATATCACGTGTTGGTGG
ICP4	For	CGTGTTTTCCGGCATGTG
	Rev	TCCCATACCAATCCTCATCCA
	Probe	FAM-CCCCCACCAGGTGCAGGCA-TAM
iNOS	For	GAGTGGTTTAAGGAGTTGGATCTGA
	Rev	TTCCAGACCTCCCACCTCAA
	Probe	FAM-CTCTGCCTGCTGTTGCCAACATGC-TAM

For, forward primer; Rev, reverse primer; FAM, 6-carboxyfluorescein; TAM, TAMRA. Underlined primer sequences anneal to the pEPkan-S plasmid used as a template to amplify the mutagenesis cassette. Italic sequences indicate the FLAG tag with a GS linker. Bold sequences are MDV-specific sequences used for homologous recombination.

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
