# Peer review of "Characterization of a Novel Viral Interleukin 8 (vIL-8) Splice Variant Encoded by Marek’s Disease Virus"

_microorganisms, 2021, doi:10.3390/microorganisms9071475_

Round 1
Reviewer 1 Report
In the present study of “Characterization of a Novel Viral Interleukin 8 (vIL-8) Splice 2 Variant Encoded by Marek’s Disease Virus”, You and colleagues confirmed and characterized the expression of a new vIL-8 protein (vIL-8-E3’) containing a novel exon though alternative splicing. They also showed that this vIL-8-E3’ protein is dispensable for MDV replication and functional vIL-8 secretion. They finally confirmed that the vIL-8-E3’ knockout virus showed similar pathogenesis and oncogenesis in chicken compared with wild type virus. The study is straightforward, and the authors clearly presented their study. Although most of the function of vIL-8-E3’ is negative, they confirmed the novel splice and protein expression of vIL-8.
I have two general questions as follow.
- How is the expression level of this novel vIL-8-E3’ compared with functional vIL-8?
- The authors mutated the acceptor site (A19’) to delete the vIL-8-E3’ expression. Although there are references showed mutating the splice acceptor sites abrogated individual splice variants without affecting other gene products, however, cryptic acceptor sites would be used to produce new splice alternatives after original acceptors were mutated. A vIL-8 N-terminal Flag tagged mutant is better to be used for characterization of E3’ deletion, meanwhile, this vIL-8 N-terminal Flag tagged mutant also could answer my first question.
- The authors claimed that there is a stop codon at the end of the novel E3’. It’s better to mark the stop codon in figure 1 to avoid the understanding that this E3’ extend to E3.
Author Response
We thank reviewers for their careful evaluation of our work and provide here a point-by-point response to all questions and comments.
Reviewer 1:
In the present study of “Characterization of a Novel Viral Interleukin 8 (vIL-8) Splice 2 Variant Encoded by Marek’s Disease Virus”, You and colleagues confirmed and characterized the expression of a new vIL-8 protein (vIL-8-E3’) containing a novel exon though alternative splicing. They also showed that this vIL-8-E3’ protein is dispensable for MDV replication and functional vIL-8 secretion. They finally confirmed that the vIL-8-E3’ knockout virus showed similar pathogenesis and oncogenesis in chicken compared with wild type virus. The study is straightforward, and the authors clearly presented their study. Although most of the function of vIL-8-E3’ is negative, they confirmed the novel splice and protein expression of vIL-8.
I have two general questions as follow.
- How is the expression level of this novel vIL-8-E3’ compared with functional vIL-8?
Thanks for this great question. We and colleagues previously performed RNA-Seq analyses of infected primary B cells and CEFs. We observed that the coverage of the vIL-8 is 13-40-fold higher compared to the vIL-8-E3’. We included this information in the manuscript (line 307).
- The authors mutated the acceptor site (A19’) to delete the vIL-8-E3’ expression. Although there are references showed mutating the splice acceptor sites abrogated individual splice variants without affecting other gene products, however, cryptic acceptor sites would be used to produce new splice alternatives after original acceptors were mutated. A vIL-8 N-terminal Flag tagged mutant is better to be used for characterization of E3’ deletion, meanwhile, this vIL-8 N-terminal Flag tagged mutant also could answer my first question.
Thanks for the great suggestion. The vIL-8 N-terminal Flag tagged virus would be indeed a very informative virus. We so far did not place a tag at the N-terminus of vIL-8, as this may interfere with its localization (Palmer and Freeman, 2004; Comparative and Functional Genomics). But we will definitely follow up on this suggestion in future studies as it will take us quite some time to generate and characterize this virus.
- The authors claimed that there is a stop codon at the end of the novel E3’. It’s better to mark the stop codon in figure 1 to avoid the understanding that this E3’ extend to E3.
As suggested by the reviewer, we marked the stop codon in Figure 1 to provide a better visualization of the exons.
Reviewer 2 Report
The authors investigated the expression of a vIL-8 splice valiant, which was recently discovered in a transcriptome analysis of Marek’s disease virus-infected cells, and its potential roles in the virus replication and pathogenesis. The authors demonstrated that this splice variant is actually translated to protein by using an exon specific tag sequence. The authors also demonstrated the lack of this vIL-8 splice variant does not affect any of virus replication in vitro and in vivo, virus transmission between hosts or the viral pathogenesis by using recombinant viruses, in contrast to the one that lacks the expression of secretory vIL-8. The experimental design was scientifically sound, the results were clearly presented, and the conclusion was supported by the results.
Minor point
Lines 150-151: The authors claim that the experiments were performed in a blinded manner. What was blinded? How can it be blinded while each group contained different number of animals?
Author Response
The authors investigated the expression of a vIL-8 splice valiant, which was recently discovered in a transcriptome analysis of Marek’s disease virus-infected cells, and its potential roles in the virus replication and pathogenesis. The authors demonstrated that this splice variant is actually translated to protein by using an exon specific tag sequence. The authors also demonstrated the lack of this vIL-8 splice variant does not affect any of virus replication in vitro and in vivo, virus transmission between hosts or the viral pathogenesis by using recombinant viruses, in contrast to the one that lacks the expression of secretory vIL-8. The experimental design was scientifically sound, the results were clearly presented, and the conclusion was supported by the results.
Minor point
- Lines 150-151: The authors claim that the experiments were performed in a blinded manner. What was blinded? How can it be blinded while each group contained different number of animals?
Thanks for raising this point. The mutant groups had initially the same number of birds (n=25) and were infected in a blinded manner. One bird of the ΔMetvIL-8 died during the first 3 days due to early mortality. Our animal use agency reduced the wild type group size in our permit to 10 birds according to the 3R principle, as our infection model with wild type RB-1B is highly reproducible. Therefore, this group could not be truly blinded.
Round 2
Reviewer 1 Report
No further comments to the authors.